# Insights into Chlamydia Development and Host Cells Response

**DOI:** 10.3390/microorganisms12071302

**Published:** 2024-06-26

**Authors:** Shuaini Yang, Jiajia Zeng, Jinxi Yu, Ruoyuan Sun, Yuqing Tuo, Hong Bai

**Affiliations:** Tianjin Key Laboratory of Cellular and Molecular Immunology (The Educational Ministry of China), Tianjin Institute of Immunology, Department of Immunology, School of Basic Medical Sciences, Tianjin Medical University, Tianjin 300070, China; nier1998@163.com (S.Y.); zjiajia814@163.com (J.Z.); yujinxi1107@tmu.edu.cn (J.Y.); sunry0609@163.com (R.S.); tuoyuqing2021@163.com (Y.T.)

**Keywords:** Chlamydia, biphasic developmental cycle, nutrient acquisition and metabolism, host cell responses

## Abstract

Chlamydia infections commonly afflict both humans and animals, resulting in significant morbidity and imposing a substantial socioeconomic burden worldwide. As an obligate intracellular pathogen, Chlamydia interacts with other cell organelles to obtain necessary nutrients and establishes an intracellular niche for the development of a biphasic intracellular cycle. Eventually, the host cells undergo lysis or extrusion, releasing infectious elementary bodies and facilitating the spread of infection. This review provides insights into Chlamydia development and host cell responses, summarizing the latest research on the biphasic developmental cycle, nutrient acquisition, intracellular metabolism, host cell fates following Chlamydia invasion, prevalent diseases associated with Chlamydia infection, treatment options, and vaccine prevention strategies. A comprehensive understanding of these mechanisms will contribute to a deeper comprehension of the intricate equilibrium between Chlamydia within host cells and the progression of human disease.

## 1. Introduction

The Chlamydia phylum consists of a solitary class and order, while the Chlamydia order encompasses eight families based on sequence similarity of the 16S rRNA gene. The Chlamydiaceae family currently comprises the singular genus Chlamydia, with 15 well-known characterized species and several newly identified candidate species [1,2]. This review primarily focuses on Chlamydia species that exhibit close associations with human diseases. Chlamydia, as Gram-negative obligate intracellular bacteria, rely on host-derived nutrients to support their growth, development, reproduction, and successful infection while concurrently modulating the metabolic mechanisms and physiological processes of host cells. Different kinds of pathogenic Chlamydia infect different hosts and cause different types of diseases. Three Chlamydia species are pathogenic to humans: Chlamydia trachomatis (*C. trachomatis*), Chlamydia psittaci (*C. psittaci*), and Chlamydia pneumoniae (*C. pneumoniae*). *C. trachomatis* is the leading bacterial cause of sexually-transmitted infections and trachoma worldwide. *C. psittaci* primarily affects psittacine birds and can be transmitted to humans causing psittacosis. *C. pneumoniae* is a major cause of respiratory tract diseases (community-acquired atypical pneumonia, bronchitis, pharyngitis, and sinusitis).

In order to successfully develop anti-Chlamydia vaccines, it is crucial to gain a comprehensive understanding of Chlamydia’s growth, development, metabolism, and host anti-Chlamydia immune response. The Chlamydiaceae family undergoes a characteristic biphasic developmental cycle, including a crucial morphological transition from elementary bodies (EBs) to reticulate bodies (RBs) upon entering the host cells for propagation, and then differentiating back into EBs to generate infectious progeny [3,4]. Chlamydia can enter a reversible persistent state when limiting nutrients such as lipids and amino acids or applying antibiotics such as penicillin, resulting in aberrant bodies, incomplete development, ongoing metabolism, and altered gene expression. After the elimination of the factors that promote persistence, Chlamydia has the ability to reactivate and undergo a phase of rapid development. The death of host cells has long been considered the ultimate phase in the Chlamydia infection cycle, and the manner in which EBs are released affects the degree of tissue damage and inflammation at the site of infection. These factors subsequently affect the potency of the immune response, ultimately regulating the survival and transmission potential of Chlamydia.

## 2. The Biphasic Reproductive Cycle of Chlamydia

### 2.1. The Genomic Characteristics of Chlamydia

The diversity of *C. trachomatis* is primarily characterized by serotyping or genotyping based on the *ompA* gene sequence, which encodes the major outer membrane protein (MOMP), a major surface antigen, and through analysis of the 16S rRNA gene. A growing body of research has demonstrated that the analysis of Chlamydia genomics can provide a more comprehensive understanding of its distinct life cycle, host–pathogen interactions, and genetic variations in Chlamydia strains associated with diverse host and tissue chemotaxis [5,6].

The analysis of the *C. trachomatis* genome revealed that the Chlamydia genes possess a minimal gene set essential for DNA replication, transcription, and translation without any redundant genes, while being abundant in DNA repair and recombination genes [6]. The recombinant genes are predominantly localized in close proximity to well-known virulence genes, including polymorphic membrane proteins (Pmps) and membrane proteins (tarps) [6]. In terms of metabolism, most Chlamydia species exhibit similar metabolic capacities, which are significantly diminished. The Chlamydia genome not only contains genes associated with crucial enzymes in aerobic respiration, but also harbors protein-coding genes resembling virulence factors found in other bacteria. Furthermore, it contains a substantial number of Chlamydia-specific genes, albeit their functional characterization remains elusive [7]. The genome of the Chlamydia transporter contains a highly developed and complete Type III secretion system (T3SS), which is universally present in Chlamydiae for acquiring indispensable nutrients from the host cell and evading its innate defense mechanisms [8]. The Chlamydia Plasticity Region (PZ) is universally present in all Chlamydia genomes, serving as an indispensable component with distinct variations across species. The PZ range spans from 18 to 81 kb and encompasses a diverse array of virulence factors, including cytotoxins, MAC/perforin, and phospholipase D enzymes [6]. The genomic analysis findings provide valuable support for the advancement of Chlamydia vaccines and the identification of appropriate therapeutic targets.

### 2.2. The Reproduction Mode of Chlamydia

Undoubtedly, all members of Chlamydia exhibit a distinct biphasic developmental cycle, alternating between two distinct morphological forms: the small (~0.3 μm), extracellular, infectious, nonreplicative elementary bodies (EBs) and the larger (~1 μm), intracellular, non-infectious, replicative reticulate bodies (RBs). However, the reproductive mechanism of Chlamydia has been extensively debated in recent years. Previous studies have indicated that RBs can be split either equally or unequally. Jennifer K. Lee and colleagues use quantitative three-dimensional electron microscopy to show that *C. trachomatis* RBs divide by binary fission, in which a parent bacterial cell gives rise to two daughter cells of equal size and volume [9]. However, Natalie Sturd and Elizabeth A. Rucks emphasize clearly that *C. trachomatis* divides by a polarized budding mechanism, as opposed to binary fission, in which daughter cells initially expand out of one pole of RBs [10,11]. Despite the initial asymmetry, the Chlamydia division process eventually produces two equally sized daughter cells, suggesting the presence of a potential cell size regulation mechanism [11]. The latest research has challenged the conventional binary classification of Chlamydia and introduced the concept of polarized budding division, thereby offering a novel perspective for investigating Chlamydia reproduction.

### 2.3. The Morphological and Physiological Differences of EBs and RBs

EBs’ membranes are composed of intra- and intermolecular disulphide bonds, primarily consisting of the cysteine-rich proteins OmcA and OmcB and the major outer membrane protein (MOMP). These cysteine-rich outer membrane proteins provide EBs with the necessary structural rigidity to maintain cellular integrity in an extracellular environment, thereby contributing to their relatively high resistance against harsh extracellular conditions [12]. Quantitative proteomics analysis reveals that EBs possess a wealth of proteins essential for central metabolism and glucose catabolism [13]. Furthermore, a study conducted in an axenic system indicates that EBs exhibit high metabolic and biosynthetic activities, relying on D-glucose-6-phosphate as their energy source [14]. This energy source may fuel a burst of metabolic activity upon entry into host cells and drive the differentiation into RBs.

Upon coming into contact with a host cell, EBs are rapidly internalized into a vacuole (termed an ‘inclusion’) and utilize T3SS to secrete a wide assortment of effector proteins into the cell cytoplasm [11]. Within 8–12 h, EBs differentiate into metabolically active RBs that replicate over a period of 24–48 h (Figure 1). The decrease in the outer membrane complex protein and COMC disulfide bond in RBs enhances membrane fluidity while increasing the porin activity of MOMP. These changes contribute to the replication of RBs. RBs highly express proteins involved in ATP generation, protein synthesis, and nutrient transport, such as nucleotide transporters, V-type ATP synthases, and ribosomal proteins [13]. Therefore, RBs have the ability to depend on ATP cleared from the host as a source of energy and participate in acquiring nutrients and replicating [15].

### 2.4. The Biphasic Developmental Trajectory of Chlamydia

#### 2.4.1. EBs Adhesion

The effective adhesion to host cells plays a crucial role in the invasion and survival of Chlamydia (Figure 1). The attachment of Chlamydia involves two steps: low-affinity electrostatic interaction with heparin sulfate proteoglycan (HSPGs) and then high-affinity binding to host cell receptors. This binding triggers the effector proteins inject into the host cytoplasm and facilitates Chlamydia uptake through T3SS [4].

Early research primarily focused on the MOMP as an adhesin [13]. Recent findings have revealed that the *C. trachomatis* polymorphic membrane proteins (Pmps) family, including Pmp21 (also known as Cpn0963) and CT017 (also known as Ctad1) function as adhesins and invasins by binding to the host cell epidermal growth factor receptor (EGFR) [16,17] and integrin-β1 receptor subunit (ITGB1), respectively. This interaction activates ERK1 and ERK2 to facilitate invasion [4,18]. Fibroblast growth factor 2 (FGF2) interacts with EBs through a dependence on HSPGs, which facilitates binding to the fibroblast growth factor receptor (FGFR). This pathway is reliant on bacterial protein synthesis and activation of the ERK1/2 signaling pathway [19]. Protein disulfide isomerase (PDI), a component of the estrogen receptor complex, plays a crucial role in facilitating the attachment and entry of Chlamydia. PDI can reduce the Chlamydia surface of the protein tightly cross-linked by disulfide bonds before entering host cells and subsequently invading host cells through host receptors associated with PDI [20]. In addition, *C. trachomatis* also uses several other receptor tyrosine kinases (RTKS), including ephrin A2 receptors (EPHA2) [21], EGFR [22], and PDGFR [23], to interact with host cells and establish inclusion.

#### 2.4.2. EBs Entry

When the EBs adhere to host cells, they deliver a range of effectors and chaperones into the cell cytoplasm (Figure 1). Pre-packaged effectors are delivered via the T3SS to initiate rearrangements of cytoskeleton and induce actin remodeling, facilitating rapid internalization of the invading Chlamydia [18] and activation of host signaling pathways [24].

Chlamydia T3SS effectors including Translocated early phosphoprotein (TepP, also known as CT875), Translocated actin-recruiting phosphoprotein (TarP, also known as CT456), CT166, Translocated membrane-associated effector A (TmeA, also known as CT694, and TmEBs. TarP is a protein with multiple domains that facilitates actin nucleation and polymerization through its globular actin/monomeric actin (G-actin) and filamentous actin (F-actin) domains, synergizing with the host ARP2/3 complex [25,26]. TmeA is a multidomain protein that disrupts the dynamics of actin by interacting with AHNAK11 [27], an actin-binding protein. This interaction occurs through its localization in the membrane and AHNAK domain. TmeA has the ability to independently activate neural Wiskott–Aldrich syndrome protein (N-WASP) for entry, regardless of TarP, or facilitate actin polymerization dependent on ARP2 and ARP3 [28,29]. The cytotoxin CT166, which is situated in the plasticity zone of *C. muridarum*, *C. trachomatis* genital strains, and *C. caviae*, exerts its effect on RHO GTPase RAC1 by means of glycosylation, potentially leading to actin depolymerization subsequent to cellular entry [15]. Host tyrosine kinases can phosphorylate TepP, which in turn recruits the eukaryotic adaptor proteins CRKI and CRKII to the inclusion. Spatially restricted activation of phosphoinositide 3-kinase (PI3K) by TepP has been demonstrated to modulate cell signaling and potentially influence membrane trafficking events downstream of TarP [30]. Both TepP and TarP, along with CT695 and CT694, utilize the chaperone Slc1 (also known as CT043) for secretion. The differential interaction between TarP and TepP with Slc1 may account for the sequential secretion of TarP before TepP [30,31]. The function of TmEBs remains unknown.

Actin polymerization is accompanied by extensive membrane remodeling, which is facilitated by host factors such as clathrin, caveolin, and cholesterol-enriched microdomains [15,32]. Mediated by clathrin or vesicles, the bacterial binding membrane begins to invaginate. The filopodia of host cells can also mediate the uptake of EBs through sorting nexin [33]. Activation of RAS-related C3 botulinum toxin substrate 1 (RAC1) recruits the actin regulators Wiskott–Aldrich syndrome protein family member 2 (WAVE2; also known as WASF2), actin-related protein 2 (ARP2), ABL interactor 1 (ABI1), and ARP3, which are necessary for actin reorganization and required for the entry of *C. trachomatis* [34]. Additionally, RHO-family GTPases, regulators of actin polymerization, are essential for internalization [12].

#### 2.4.3. EBs to RBs Conversion

Following endocytosis, multiple EBs bind and enter the same host cells to form individual inclusions, which subsequently fuse together through homotypic fusion to create a single large inclusion. Within these inclusions, EBs undergo differentiation into replicative RBs [35] (Figure 1).

Chlamydia triggers genome replication and biosynthesis of peptidoglycan precursors, resulting in the loss of infectivity while acquiring distinct phenotypic and biochemical characteristics similar to RBs. The addition of metabolites, such as glutamine and G6P, to the axenic medium leads to a remarkable transformation of EBs morphology into an intermediate form that exhibits bridging characteristics between EBs and RBs [36]. Glutamine plays a pivotal role in the biosynthesis of peptidoglycan precursor, which is predominantly synthesized during RBs division, making it a crucial nutritional signal for the transition from EBs to RBs. However, it is important to acknowledge that axenic cultures do not fully represent the intricate host cell environment within the *C. trachomatis* inclusion [35].

#### 2.4.4. RBs to EBs Transition

The transformation of fragile, exclusively intracellular larger RBs into more robust small EBs capable of surviving extracellularly is facilitated by relevant morphological and biochemical alterations (Figure 1). This differentiation process is facilitated by a small histone-like protein Hc1 (HctA), which acts as a DNA and RNA-binding protein that downregulates gene expression in vitro, leading to nucleoid condensation and higher density packed DNA within EBs compared to RBs [37]. Another notable change during the transition from RBs to EBs is the surface stabilization in EBs, achieved through the formation of disulfide bonds that cross-link cysteine-rich proteins [38].

In the early stage of Chlamydia reproduction, RBs may be in closer contact with the inclusion membrane, facilitating nutrient acquisition. Research has shown that detachment from the inclusion membrane and deactivation of the T3SS apparatus can directly impact nutrient uptake [39]. The limited availability of nutrients in the inclusion may restrict the size of RBs’ offspring and result in their conversion into the EBs state [40].

#### 2.4.5. EBs Release

The release of EBs from host cells is achieved through two mutually exclusive mechanisms: either by cell lysis, which is triggered by the activation of cysteine proteases, or by extrusion of Chlamydia vacuoles or liberation of Chlamydia within cytoplasmic fragments enclosed by cell membranes (Figure 1), both of which occur via an actin-dependent and myosin-dependent mechanism [41].

The release of EBs leads to host cell lysis, which involves permeabilization of the inclusion membrane, followed by permeabilization of the nuclear membrane and ultimately calcium-dependent plasma membrane lysis [41]. Research indicates that the type II secretion system (T2SS) effector, Chlamydia protease-like activity factor (CPAF, also known as CT858), and active cytosolic CPAF are potentially involved in host cell lysis and the release of EBs.

The extrusion pathway, resembling exocytosis, maintains the integrity of the host cell by involving membrane pinching followed by expulsion of an inclusion. Hybiske and Stephens discovered that these extrusions are vesicles filled with Chlamydia that bud off from infected cells [41]. These vesicles contain Chlamydia enclosed within an intact inclusion, which is surrounded by cytoplasm derived from the host cell. During extrusion, the infected cell can either release its entire intracellular bacterial load or retain a smaller inclusion. Eventually, these released extrusions undergo lysis to liberate EBs, enabling them to infect new host cells [38].

## 3. Nutrient Acquisition and Intracellular Metabolism of Chlamydia

### 3.1. Nutrients Acquisition

*Chlamydia* spp. lack necessary biosynthetic enzymes and even entire metabolic pathways [32,42]; thus, they rely on host cells to absorb nutrients for their own growth and reproduction. The acquisition of essential nutrients by Chlamydia inclusions involves selective redirection of transport vesicles and hijacking of intracellular organelles. This process is accomplished through various mechanisms, including the recruitment of Rab GTPases and SNAREs, translocation of organelles into the inclusion lumen, and proteolysis-mediated fragmentation of the Golgi apparatus [43].

Eukaryotic lipids, such as phosphatidylcholine, phosphatidylinositol, sphingomyelin, and cholesterol, are essential for the Chlamydia membrane. These lipids play essential roles in various processes, including homotypic fusion, replication, growth and stability of the inclusion membrane, reactivation from persistence, and RBs to EBs re-differentiation. Chlamydia also targets cholesterol as a potential membrane source during host cell colonization and manipulates host cell signaling and trafficking [44,45]. Sphingomyelin and cholesterol are obtained from either the Golgi apparatus or multivesicular bodies [46]. Lipid droplets (LDs) are storage organelles of neutral lipids that accumulate and enter Chlamydia inclusions during Chlamydia infection, which are endoplasmic reticulum derived organelles consisting of neutral lipid nuclei surrounded by phospholipids monolayer [47,48].

The availability of appropriate carbon sources and energy is essential for the production of metabolic intermediates and cellular reactions. Due to the lack of the hexokinase gene, which catalyzes the conversion of glucose into glucose 6-phosphate (G6P), Chlamydia is unable to directly metabolize host glucose. Throughout its developmental cycle, Chlamydia expresses a gene encoding a glucose 6-phosphate transporter (UhpC), as well as genes encoding key enzymes in glycolysis and the pentose phosphate pathway. This allows them to directly utilize host G6P through UhpC anti-transporter and enter metabolism. The G6P required by Chlamydia can be acquired from host cells, inclusion vacuoles, or cytoplasmic glycogen reservoirs, which can either be directly obtained from the host cytoplasm or synthesized through the introduction of glucosidine diphosphate into inclusions. Although G6P can be oxidized via either pathway, only oxidation through the glycolytic pathway would result in ATP production. In the reproductive stage of RBs, G6P is primarily utilized for LPS synthesis. RBs highly express proteins involved in ATP generation, protein synthesis, and nutrient transport, such as V-type ATP synthases, ribosomal proteins, and nucleotide transporters. This renders Chlamydia entirely dependent on host ATP as its energy source [13,49]. Once RBs convert into EBs, the primary source of energy is derived from ATP generated through the conversion of G6P to pyruvate in glycolysis. Additionally, EBs can also acquire energy by degrading glycogen in the extracellular environment [42,50].

The Chlamydia genome has been revealed to have incomplete or missing biosynthetic pathways for most amino acids, necessitating the uptake of exogenous amino acids. To facilitate this, Chlamydia possesses various amino acid transporters, including neutral amino acid transporter, amino acid anti-transporter, branch chain amino acid transporter, and a multitude of ABC transporters [43]. When competing with host cells for available amino acids, Chlamydia predominantly relies on glutamine, histidine, phenylalanine, leucine, and valine for growth [51].

Despite the ability of Chlamydia-encoded enzymes to generate ATP through substrate-level phosphorylation and synthesize CTP from UTP via CTP synthetase, these microorganisms still depend on ATP, GTP, and UTP derived from host cells for their metabolic requirements. It is worth noting that the genome of *C. trachomatis* does not contain the complete set of enzymes necessary for nucleotide biosynthesis, thus necessitating compensation by the host for the supply of nucleotides [43]. Studies have shown that *C. trachomatis* possesses at least two nucleotide transporters (Npt): Npt1 and Npt2. Specifically, Npt1 facilitates the import of ATP and the export of ADP from host cells to bacteria, while Npt2 catalyzes the proton-dependent uptake of GTP, UTP, CTP, and ATP [52]. Different Chlamydia species exhibit variations in their nucleotide metabolism capabilities [43].

### 3.2. Intracellular Metabolism

Chlamydia’s reliance on its host as a replicative niche is intricately tied to its metabolism, which in turn affects its adaptability to the environment, reproductive capabilities, and ability to function independently [53,54]. The central carbon metabolism of Chlamydia encompasses a complete glycolytic pathway, the pentose phosphate pathway (PPP), and a partial citrate (TCA) cycle [55]. Studies have also confirmed that PEP carboxykinase (PckA), which converts oxaloacetate into PEP, utilizes glutamine/glutamate or dicarboxylic acids from the TCA cycle as additional carbon sources [55]. Indeed, *C. trachomatis* growth in cell culture was feasible with malate, glutamate, 2-oxoglutarate or oxaloacetate as major carbon sources, although chlamydial replication rate decreased significantly compared to when glucose was used as the primary carbon source [56]. Each of these carbon sources plays stage-specific roles.

Chlamydiaceae lack citric acid synthetase (GltA), aconitase (Acn), and isocitric acid dehydrogenase (Icd), resulting in an incomplete TCA cycle [57,58]. Moreover, the absence of pyruvate carboxylase prevents the conversion of pyruvate to oxaloacetate, and the chlamydial genome lacks a gene homologous to an oxoglutarate synthase, which explains the unidirectional formation of succinate by 2-oxoglutarate dehydrogenase (Suc) [54]. *C. trachomatis* possesses all the necessary enzymes for the PPP, enabling it to produce both NADPH and pentose phosphates required for their synthesis. Additionally, it has a complete gluconeogenesis pathway and the essential enzymes and capacity for glycogen synthesis and degradation [14,59]. Furthermore, Chlamydia is capable of encoding an enzyme that facilitates the conversion of glucose-6-phosphate into pyruvate through glycolysis, allowing them to generate ATP via substrate-level phosphorylation mediated by phosphoglycerate kinase and pyruvate kinase enzymes. However, the connections between glycolysis and the partial TCA cycle appear to be unidirectional. PckA only facilitates the conversion of oxaloacetate into PEP. Genome sequences indicate that acetyl-CoA derived from the end product of glycolysis and fatty acid degradation is unable to enter the TCA cycle.

Genome analysis shows that Chlamydiaceae has a complete and minimal respiratory chain that can produce ATP through oxidative phosphorylation. The chain includes Na^+^-translocating NADH dehydrogenase (Nqr, complex I), succinate dehydrogenase (SdhA-C, complex II), cytochrome bd oxidase (CydAB, complex IV), and a V-type ATPase (complex V) [14]. The potential presence of a cytochrome bd-like oxidase, which typically demonstrates a high oxygen affinity, suggests that Chlamydiae might face limited oxygen levels while growing inside cells and consequently exhibit characteristics of a microaerophilic lifestyle [60,61]. This respiratory chain facilitates the formation of oxidized forms of NADH and FADH2, which are subsequently reduced during glycolysis and the TCA cycle. Additionally, it enables bacteria to build an electrochemical membrane potential on the plasma membrane to drive transport [62].

## 4. Pro-Death Effects of Chlamydia

### 4.1. Exit of Host Cells

In research conducted by Hybiske and Stephens, it was observed that the late-stage host cell death occurs in a sequential pattern from inside to outside. This process is initiated by the rupture of the inclusion membrane, followed by subsequent ruptures of other intracellular structures like the nuclear envelope. Eventually, it culminates in the rupture of the host plasma membrane. The inclusion rupture is effectively inhibited and the duration of host cell death is significantly extended by E64, a potent cysteine protease inhibitor. This finding supports the hypothesis that bacteria play a role in inducing inclusion rupture. Further support came from another study that revealed a delay or obstruction in the natural release of *C. trachomatis* L2 from HeLa cells when chloramphenicol, an inhibitor of bacterial protein synthesis, was administered during the later stages of infection. Furthermore, the inclusion and/or host plasma membrane lysis is influenced by factors such as CPAF, PGP4, and CT153, along with the involvement of the protease enzyme [63].

To investigate the release of Chlamydia from infected cells, Kerr et al. employed multiphoton ablation to selectively disrupt the chlamydial inclusions within the host cells. They observed that in the event of inclusion rupture ablation at any given moment during the infection, it results in subsequent rupture of the host’s plasma membrane and could potentially expedite the growth of the bacteria. As the chloramphenicol treatment and inhibition of CPAF had no impact on ablation-induced cell death, it can be inferred that the plasma membrane rupture is driven by the host cell. Additionally, it is suggested that host calpains might contribute to inclusion rupture, thereby further promoting host cell membrane disruption, as indicated by a delay in inclusion rupture when calpain inhibitors are present [64].

### 4.2. Cytotoxicity of Chlamydia

The term ‘multiplication-independent immediate toxicity’, termed by Moulder et al., refers to the cytotoxic effects observed in cultured eukaryotic cells after exposure to high multiplicities of infection (MOIs) of EBs. In the early 1940s, Rake et al. demonstrated that intravenous administration of Chlamydia (*C. trachomatis*, *C. muridarum,* or *C. psittaci*) caused toxemia and rapid death or typical signs of infection such as ruffled fur, hunched back, and weight loss in mice within 4–24 h post-infection [65]. Injection of inactivated Chlamydia or corresponding antiserum with Chlamydia, or pretreatment with Chlamydia suspensions prior to infection, could inactivate the toxin and protect mice from death [65]. These results suggest that infectious EBs may produce toxins, particularly the D serovar and the L2c variant of the L2 serovar of *C. trachomatis*, highlighting the significant role of EBs toxin-like activity as a virulence factor in Chlamydia pathogenesis [66,67].

High doses of Chlamydia induce rapid cell death in cultured cells (murine fibroblasts, HeLa cells, and murine macrophages), which can be attributed to direct physical damage. This phenomenon is commonly referred to as immediate cytotoxicity. Immediate cytotoxicity exhibits a necrotic nature with varying degrees of potency. The toxicity of high doses of *C. psittaci* for L cells was not reduced by inhibitors targeting bacterial protein synthesis and transcription [68]. When cells were exposed to lower doses of Chlamydia (10–100 ID_50_ per cell), both multiplication-independent and multiplication-dependent toxicity were observed. However, when the doses dropped below 10 ID_50_ per cell, the toxicity that was not related to bacterial replication disappeared, and the induction of host cell death became reliant on both entry into the cell and intracellular bacterial replication. Additionally, host cell damage was only observed at 48–72 h post-infection in response to these low doses, indicating the occurrence of late-stage host cell death [69].

The impact of Chlamydia extends beyond the viability and proliferation of infected cells, also affecting the fate of uninfected cells. Research has revealed that neighboring cells without inclusion exhibit a significant elevation in apoptosis levels compared to Chlamydia-infected epithelial cells [70]. Furthermore, *C. trachomatis*-infected macrophages release TNFα, which plays a crucial role in inducing apoptosis of T cells [71]. In addition to its potential for causing tissue damage, inflammation, and post-infection sequelae, Chlamydia-induced cell death in neighboring cells may deplete immune cells and impede anti-chlamydial immune responses.

## 5. Host Cells Defend against Chlamydia

### 5.1. Cell Fates

#### 5.1.1. Apoptosis [72]

As early as 1998, Ojcius and colleagues used various methods to demonstrate that *C. caviae* can induce apoptosis in Hela cells and murine macrophages [73]. Subsequently, numerous scientists have demonstrated this phenomenon using fluorescence ANNEXIN 5, and in certain studies, microscopic assays have also confirmed the occurrence of apoptosis-like characteristics in cells containing Chlamydia inclusions (Figure 2). Furthermore, DNA fragmentation and nuclear condensation were observed in cells infected with *C. muridarum*, *C. psittaci*, or *C. trachomatis* L2 [70].

The two major pro-apoptotic stimuli, STS and TNF/CHX, trigger apoptosis through distinct signaling pathways: the intrinsic pathway and the extrinsic pathway of apoptosis, respectively. In the intrinsic pathway, MOMP is mediated by pro-apoptotic BCL-2 family proteins BAX and BAK [74]. This process is tightly regulated by both pro-apoptotic BH3-only proteins and anti-apoptotic BCL-2 family proteins [31,75], leading to the release of cytochrome c and activation of the initiator caspase CASP9 [76]. Subsequently, CASP9 activates effector caspases (CASP3 and CASP7), which triggers the process of cellular degradation [32]. In the extrinsic pathway, the activation of initiator caspase CASP8 can be initiated by the formation of a signaling complex through the interaction with death receptors like TNF-α receptor [35]. In most cases, induction of MOMP via CASP8-dependent pro-death signaling amplification is required for cell death induction, although in some cell types, CASP8 has the ability to directly activate apoptotic effector caspases [36].

#### 5.1.2. Necroptosis

By utilizing time-lapse video microscopy to monitor the fate of infected cells and analyzing host plasma membrane integrity as well as caspase activity, Barbara S. Sixt et al. have demonstrated that Chlamydia-infected cells do not evade death under pro-apoptotic conditions, but rather undergo an atypical form of necrosis. This necroptosis is partly impacted by the presence of bacteria and the cells that are infected undergo necrosis, which is characterized by an abrupt rupture of the plasma membrane of the host cell and the subsequent release of cellular contents (Figure 2). This process does not involve the activation of apoptotic effector caspases or exhibit morphological features typically associated with apoptosis [77].

#### 5.1.3. Autophagy

The induction of autophagy by Chlamydia depends on the specific species and cell line involved (Figure 2). Autophagy is significantly increased during the replicative stages of *C. trachomatis* infection in LGV disease. Furthermore, both *C. pneumoniae* and *C. trachomatis* growth have been found to be inhibited when autophagy inhibitors, such as 3-methyladenine (3-MA) and bafilomycin A1 (BafA1), are administered. Interestingly, the absence of autophagy-related proteins Atg5^−/−^ or Irga6^−/−^ in mouse embryonic fibroblasts (MEFs) leads to an intriguing increase in chlamydial growth when stimulated by IFNγ, as compared to normal MEFs [78,79]. This suggests that these proteins play a vital role in facilitating autophagy-mediated defense against *C. trachomatis* infection [79]. By evading elimination from the host’s innate and adaptive immune systems and exploiting autophagy processes, *C. trachomatis* establishes a persistent infection [80].

#### 5.1.4. Pyroptosis [72] (Figure 2)

Multiple studies have demonstrated that *Chlamydia* spp. possess the ability to initiate canonical inflammasomes, specifically targeting the PYD, NACHT, and LRR domains containing protein 3 (NLRP3) and absent in melanoma 2 (AIM2) inflammasomes, typically within a short period following infection. Consequently, this infection has the potential to induce CASP1 activation and subsequent secretion of pro-inflammatory cytokines IL-1β and IL-18. However, the main focus of these studies was to examine the relationship between *Chlamydia* spp. and human or murine monocytes or macrophages. Limited research has indicated the presence of inflammasome activation in epithelial cells, although it exhibits a delayed response. Compared with *C. trachomatis* (D or L2), *C. caviae*, and *C. muridarum*, *C. pneumoniae* showed a stronger ability to induce inflammasome and CASP1 activation. There is evidence indicating that the non-canonical inflammasome pathway can be triggered by both *C. trachomatis* and *C. muridarum*, resulting in pyroptotic cell death in infected macrophages derived from mouse bone marrow (BMDMs), with involvement of CASP1 and CASP11. [81,82]. Moreover, several studies have indicated that the deficiency or suppression of CASP1 within infected host cells may adversely affect the intracellular proliferation of Chlamydia [83,84,85].

### 5.2. Anti-Death Strategies of Host Cells

#### 5.2.1. The Pro-Survival Pathways

Chlamydia can activate the pro-survival pathways of host cells and anti-apoptotic pathways. The intracellular survival signaling pathways activated by Chlamydia infection encompass the polo-like kinase 1/3-phosphoinositide-dependent protein kinase 1/Myc proto-oncogene (PLK1/PDPK1/MYC) signaling pathway, the Raf/MEK/ERK mitogen-activated protein kinase (MAPK) pathway, and the phosphoinositide 3-kinase/protein kinase B (PI3K/AKT) pathway [63,64,86,87].

*C. trachomatis* activates the PI3K pathway by binding to EPHA2, leading to internalization of these receptors with EBs. This internalization continues to induce persistent survival signals crucial for bacterial replication [88]. The upregulation of ERK increases the level of EPHA2, resulting in the activation of a feed-forward loop involved in host survival [88]. The ERK-MAPK signaling pathway plays a crucial role in regulating bacterial nutrient acquisition, expression of anti-apoptotic factors, and synthesis of pro-inflammatory cytokines [89,90,91]. Targeted inhibition of these signaling pathways can enhance the susceptibility of infected cells to staurosporine (STS)- and Granzyme B-induced apoptosis [91,92]. Additionally, early ERK activation has been linked to the Chlamydia effector protein TARP phosphorylating SHC1 and activating the MEK/ERK pathway [93]. Recent studies have also suggested that the secretion of the plasmid-encoded Chlamydia protein PGP3 may inhibit apoptosis by inducing the PI3K/AKT and ERK signaling pathways [94,95,96]. In the case of *C. psittaci*, infected HeLa cells activate the ERK pathway and U0126 sensitizes infected cells to STS-induced apoptosis [97]. Finally, the Inc Cpn1027 from *C. pneumoniae* interacts with cytoplasmic activation/proliferation-associated protein 2 (CAPRIN2) and glycogen synthase kinase 3β (GSK3β), both members of the β-catenin–WNT pathway, facilitating the transcriptional activation of pro-survival genes [4].

The signaling pathway through which Chlamydia induces host cell survival is influenced by the species of Chlamydia and the type of infected cell. *C. pneumoniae* induces NFκB activation in human epithelial cells and human mononuclear cell lines Mono Mac 6, and the use of NFκB nuclear translocation inhibitors (CAPE) and siRNA-mediated depletion of the P65 subunit of NFκB can increase the sensitivity of infectious cells to TNFα/CHX- and STS-induced apoptosis [98], suggesting that NFκB activation during infection is crucial for host cell survival; as such, inhibiting it induced apoptosis in these cells [99,100]. However, the infection of human epithelial cells with *C. trachomatis* did not result in NFκB activation, and inhibiting NFκB activation in these cells did not sensitize to STS- or TNFα/CHX-induced apoptosis. In addition, the activation of the Janus kinase/signal transducer and activator of the transcription protein 3 (JAK/STAT3) pathway has been proposed as a contributing factor to the development of resistance against apoptosis during infection with this particular species [101].

#### 5.2.2. The Anti-Apoptosis Pathways

The anti-apoptotic effect of Chlamydia was initially acknowledged twenty years ago [15]. At the cellular level, host cell anti-apoptotic activity predominates during early and middle Chlamydia development, while host cells produce, including inhibiting or inducing apoptosis and other forms of cell death, during late Chlamydia development, providing an exit route for infected cells and facilitating the spread of infection [4,102,103]. At the tissue level, apoptosis is an immune escape mechanism that induces non-inflammatory death of infected cells and also limits adaptive immunity by inducing T cell apoptosis.

Host cells infected with *Chlamydia* spp. exhibit cell-autonomous resistance to the stimuli of intrinsic and extrinsic apoptosis [87]. *Chlamydia* spp. has the ability to impede intrinsic apoptosis via diverse mechanisms, including the ubiquitination of the tumor suppressor p53 by MDM2 followed by its degradation through proteasomes [63,64], the sequestration of pro-apoptotic protein kinase Cδ (PKCδ) or BCL-2-associated agonist of cell death (BAD) on the inclusion membrane via diacylglcerol or 14-3-3β-binding Incs [57], the degradation of pro-apoptotic proteins [104,105], and blockade of mitochondrial cytochrome c release, as well as upregulation or stabilization of anti-apoptotic proteins such as myeloid leukaemia cell differentiation protein 1 (MCL1), cIAP2 (also known as BIRC3), or BAG family molecular chaperone regulator 1 (BAG1) [15,95].

*C. trachomatis* hinders extrinsic apoptosis by blocking the activation of caspase 8 through cellular FLICE-like inhibitory protein, which acts as a master regulator [106] and inhibits formation of apoptosomes [107,108]. Studies have also shown that inhibition of caspase-3 activity in *C. trachomatis* and *C. pneumoniae* early infections can also be resistant to staurosporine-induced apoptosis. Additionally, a prominent mechanism for preventing cell death is CPAF-mediated degradation of the pro-apoptotic BH3-only proteins Bad, Bim, and Puma. This process leads to decreased activity of Bax and Bak, blocking the release of cytochrome c, as well as an upregulation of IAP2 stability and the pro-survival factor Mcl-1 during infection [86,104,105].

### 5.3. Anti-Chlamydia Immune Response of Host

#### 5.3.1. IFNgama (IFNγ)

During the initial 24 h period of Chlamydia infection, bacteria undergo rapid proliferation and are recognized by cellular pattern-recognition receptors such as TLR2, TLR3, NOD1, cGAS, and STING. Subsequently (days 1–3 post-infection), this recognition triggers the production of IFNγ, primarily by natural killer (NK) cells, innate lymphoid cells (ILCs), and γδT cells [109], providing sufficient stimulation to infected cells for clearing the infection and preventing dissemination. After the initial week following infection, adaptive immunity takes on the responsibility of producing IFNγ. Th1 cells are primarily responsible for generating IFNγ, with supplementary assistance from CD8^+^ T cells. The lymphocytes persist at the infectious site for several weeks and exhibit spontaneous clearance of infection within a period of 28 to 45 days [109]. Innate IFNγ plays a crucial role in limiting early *C. muridarum* dissemination, while Th1-derived IFNγ is essential for eradicating established Chlamydia infections [109].

The mechanism of IFNγ resistance to Chlamydia is as follows: (1) Upregulation of inducible nitric oxide synthase (iNOS) stimulates the synthesis of nitric oxide (NO). Although the role of NO in clearing Chlamydia remains unclear, it has been observed to damage bacterial DNA and exhibit cytotoxic effects [110]. Studies have shown that iNOS-deficient mice experience exacerbated pathological outcomes during genital tract *C. trachomatis* infection, while the growth of *C. trachomatis* is reduced in iNOS knockout mouse lung fibroblasts [111]. (2) Activation of indoleamine 2,3-dioxygenase (IDO) converts tryptophan into N-formyl kynurenine. Moreover, IDO inhibits the production of other intracellular cytokines and potentially induces apoptosis. Since *C. trachomatis* cannot synthesize tryptophan and must rely on host acquisition, the presence of this enzyme results in a depletion of host cell tryptophan, thereby inhibiting RBs growth [112]. The presence of kynurenine restricts the proliferation of CD4^+^ T cells and induces the differentiation of regulatory T cells, thereby leading to a reduction in IFNγ production and facilitating the reactivation of *C. trachomatis* [113]. The impact of tryptophan depletion on *C. trachomatis* has two aspects: Firstly, it triggers the elimination of RBs. Secondly, it restricts the proliferation of RBs by halting the synthesis of structural proteins, membrane proteins, and lipopolysaccharides. Once IFNγ is removed or IDO expression decreases, their reproductive capacity can be restored [114].

#### 5.3.2. Cell-Autonomous Immunity

Cell-autonomous immunity refers to the ability of an individual cell to independently detect, contain, and eliminate pathogens that invade the cell by inducing cellular defense responses [115]. The cell-autonomous immune responses can be categorized functionally into two groups: pathogen detection through pattern recognition receptors (PRRs) and execution of antimicrobial effector functions [115]. This is central to the pathogenesis and immunogenicity of *C. trachomatis* infections [116]. Cell-autonomous immunity against intracellular pathogens, such as Chlamydia, is partially mediated by the immunity-related GTPase (IRG) protein family, which localizes to Chlamydia inclusions and initiates lytic destruction of targeted inclusions [109]. The activation of PRRs leads to the establishment of a highly pro-inflammatory microenvironment surrounding *C. trachomatis*-infected cells, thereby inducing cell-autonomous host defense in both infected and non-infected bystander cells [115]. The infected epithelium elicits cell-autonomous immune responses, undergoes cellular alterations, and releases pro-inflammatory cytokines that recruit and activate innate immune cells including neutrophils, macrophages, dendritic cells, and potentially tissue-resident lymphoid cells to combat *C. trachomatis* as well as other intracellular pathogens [116].

#### 5.3.3. Innate Immune Response

Natural killer cells (NK cells) promptly migrate to the infected tissue during Chlamydia infection and serve as the primary early producers of IFNγ, which plays a pivotal role in shaping the adaptive immune response [113]. NK cells have the ability to eliminate *C. trachomatis*-infected epithelial cells in order to impede pathogen proliferation. Dendritic cells (DCs) are a typical antigen-presenting cell. Immature DCs secrete pro-inflammatory cytokines, such as IL-6, TNF-α, CCR7, CXCL10, IL-1α and IL-12, following the phagocytosis of *C. muridarum* to promote their own maturation and antigen presentation. DCs in vivo degrade Chlamydia and present it to naive CD4^+^ T lymphocytes via MHC II molecules to drive T cells’ activation and differentiation into Th1 subsets for the initiation of cellular immune response [117]. Macrophages migrate to the site of infection to phagocytose pathogens and produce pro-inflammatory cytokines [109]. Their clearance of *C. muridarum* is associated with autophagy, which enhances their antigen-presenting ability and subsequently boosts the T cell response. Studies have demonstrated that IFNγ can enhance macrophage autophagy and upregulate MHC II molecules [118]. Neutrophils are recruited in the early stage of Chlamydia infection and can effectively eliminate Chlamydia through complete phagocytosis of inclusion bodies. However, the extent of its protective effect may vary depending on the species of Chlamydia [109].

#### 5.3.4. Adaptive Immune Response

As the infection progresses, antigen-presenting cells initiate adaptive immunity, resulting in the production of anti-Chlamydia antibodies by B cells and the infiltration of Chlamydia-specific CD4^+^ T and CD8^+^ T cells into infected tissue. The involvement of the adaptive immune system is crucial for facilitating Chlamydia clearance and providing protection against reinfection.

Despite being obligate intracellular bacteria, Chlamydia can induce humoral immunity through B cell activation, leading to the production of antibodies. Although B cells alone may not provide protection against primary Chlamydia genital infection, they are capable of preventing reinfection by generating Chlamydia-specific IgA in genital secretions [109] and inhibiting bacterial dissemination to distant tissues [113,119].

A large amount of evidence has demonstrated that naive CD4^+^ T cells differentiate and develop into IFNγ-secreting Th1 cells under the action of IL-12, which is the main adaptive cell type for defense and clearance of primary Chlamydia infection [109,113]. In mouse models, there is little evidence to support the protective role of other T cell populations in primary Chlamydia infection. Th2 cells aggravate Chlamydia infection by inhibiting the protective Th1 response, and an increased Th2 response is associated with a high incidence of hydrosalpinx in il-12-deficient mice. IL-17 knockout mice infected with *C. muridarum* showed reduced bacterial shedding and decreased infiltration of neutrophil and macrophage at the infection site. Another study demonstrated that IL-17 receptor knockout mice exhibited partially impaired Th1 immune function during *C. muridarum* infection, indicating an indirect role for Th17 cells in supporting Th1 responses [113].

Although CD8^+^ T cells are capable of recognizing and eliminating target cells expressing *C. trachomatis* proteins Cap1 and CrpA, as well as secreting IFNγ to resolve Chlamydia infection, studies have demonstrated that the absence of CD8 or perforin-deficient mice does not affect the clearance rate of the infection compared to wild-type mice, indicating their nonessential role in clearing Chlamydia in mice [113]. In the *C. muridarum* mouse model, CD8^+^ T cells are neither necessary nor sufficient for protection against primary or secondary infection; however, they do contribute to immunopathology associated with the infection. Conversely, in a non-human primate model, CD8^+^ T cells play a crucial role in protective immunity, since depletion of these cells after immunization with live-attenuated *C. trachomatis* significantly diminishes the vaccine’s protective effect [119].

## 6. The Prevailing Disease, Treatment, and Prevention of Chlamydia

### 6.1. The Prevailing Disease of Chlamydia

#### 6.1.1. *C. trachomatis* and Trachoma

*C. trachomatis*, a bacterium responsible for causing trachoma and reproductive tract infections, is classified into different serotypes based on the antigenic properties of MOMP. Serotypes A-C are capable of inducing trachoma, a prevalent infectious disease that leads to global blindness [120]. The main symptoms include recurrent conjunctival infections during childhood, which subsequently result in chronic inflammation, conjunctival scarring, entropion (eyelid varus), trichiasis caused by eyelashes rubbing against the cornea, and ultimately adult-onset blindness. Active trachoma is characterized by a persistent immune response to Chlamydia antigen that leads to intense inflammation of the cystic conjunctiva [121,122,123]. To address this significant public health challenge, the GET2020 Alliance recommends implementing a SAFE strategy: Surgery to correct trichiasis; Antibiotics (oral azithromycin or topical tetracycline) to treat Chlamydial infection; Facial cleanliness; and Environmental improvements to reduce transmission. Additionally, the World Health Organization also advocates for mass drug administration (MDA) when the prevalence of active trachoma among children aged 1–9 years reaches 10% or higher [121].

#### 6.1.2. *C. psittaci* and Psittacosis

Psittacosis is a zoonotic infection caused by *C. psittaci*, primarily transmitted to humans from birds through inhalation of aerosolized organisms from dried feces or respiratory secretions, or direct contact [124]. The incubation period lasts 5 to 14 days and the infection can range from asymptomatic to mild disease, with severe cases being life-threatening during pregnancy, especially in the second or third trimester [124]. Clinical symptoms of psittacosis include high fever, chills, headache, myalgia, non-productive cough, and respiratory distress [120]. Severe cases can lead to multiple organ dysfunction syndrome (MODS), pneumonia, and respiratory failure [125]. The primary treatment for psittacosis is tetracycline antibiotics, and commonly prescribed options include doxycycline or minocycline. Mild to moderate infections can be effectively managed through oral administration, while severe cases may necessitate intravenous doxycycline. Symptoms typically show improvement within 24–48 h following treatment, and in order to prevent recurrence, it is recommended to administer antibiotics for a minimum duration of 14 to 21 days. Macrolides, such as azithromycin, roxithromycin, and erythromycin, are suitable alternatives, especially for pregnant women and children. Although fluoroquinolones like moxifloxacin and gatifloxacin exhibit favorable in vitro activity against *C. psittaci*, their clinical efficacy is less established and they are generally not recommended as first-line agents [125,126].

#### 6.1.3. *C. pneumoniae* and Pneumonia

*C. pneumoniae* is an obligate intracellular pathogen transmitted via aerosols and is a common cause of respiratory diseases, mainly manifesting as pneumonia and bronchitis, responsible for 10% of community-acquired pneumonia and 5% of bronchitis, pharyngitis, and sinusitis [127]. *C. pneumoniae* infection is typically mild, presenting with nonspecific symptoms like fever, cough, and shortness of breath, with most cases being asymptomatic. Severe cases of meningoencephalitis, Guillain–Barré syndrome, myocarditis, and endocarditis [120]. The primary treatment for community-acquired bacterial pneumonia (CABP) caused by *C. pneumoniae* includes antibiotics like lefamulin, a novel pleuromutilin that is available intravenously and orally and is an alternative to fluoroquinolones and other common antibiotics [128].

### 6.2. Anti-Chlamydia Drugs

#### 6.2.1. Tetracycline Antibiotics [129,130]

Tetracyclines are the preferred antibiotics for clinical treatment of Chlamydia infection due to their excellent therapeutic efficacy against *C. pneumoniae*-induced pneumonia, *C. trachomatis*-induced nongonococcal urethritis and cervicitis, venereal lymphogranuloma, and inclusion body conjunctivitis and trachoma, as well as *C. psittacosis*-induced psittacosis, regardless of oral or topical administration. Doxycycline can be considered as the drug of choice [131]. The bacteriostatic mechanism of tetracycline drugs is to inhibit the interaction between aminoacyl tRNA and ribosome, thereby impeding the synthesis of crucial bacterial proteins [132]. Upon entering the cell, these drugs specifically bind to the 30S subunit of the ribosome at the A site, obstructing tRNA entry into the mRNA-ribosome complex and consequently inhibiting peptide chain elongation. Ultimately, this disrupts protein synthesis in Chlamydia. Furthermore, tetracycline drugs also increase membrane permeability in Chlamydia, causing leakage of intracellular nucleotides and other substances that eventually lead to Chlamydia death.

#### 6.2.2. Macrolide Antibiotics [133,134]

Macrolide antibiotics, including azithromycin, erythromycin, and roxithromycin, are considered the first-line drugs for treating Chlamydia infection. Among them, azithromycin exhibits the strongest efficacy against Chlamydia infection. Mechanistically, macrolide antibiotics bind to the region between the transpeptidase center of the 50S ribosomal subunit and the peptide output channel. This binding effectively obstructs protein synthesis by physically blocking the channel and impeding peptide extension [132].

#### 6.2.3. Quinolone Antibiotics [132]

Quinolones are the second-line treatment for Chlamydia infection. The effect of quinolone antibiotics on bacteria (including Chlamydia) primarily is mainly manifested in the inhibition of DNA gyrase and topoisomerase IV, thus affecting DNA replication, transcription, and expression. DNA gyrase consists of two subunits, α and β, encoded by *gyrA* and *gyrB* genes, respectively. Quinolones act on the DNA gyrase α subunit to form a drug–DNase complex, inhibit the opening and closing of DNA chains, and hinder the synthesis of bacterial DNA, thus playing a bactericidal role [135,136]. Topoisomerase IV, a tetramer, consists of a C subunit responsible for DNA strand breakage and reattachment, and an E subunit that catalyzes ATP hydrolysis and forward chain shift. This enzymatic complex plays a crucial role in the late-stage separation process of sister chromosomes during DNA replication. The c subunit is encoded by the *parC* gene. Quinolones exert their bactericidal effect by inhibiting topoisomerase IV activity, thereby impeding DNA replication. Additionally, quinolones can induce bacterial self-dissolution by changing the composition of peptidoglycan in bacterial cell wall. Furthermore, they can also trigger SOS repair of bacterial DNA, thereby enabling the drug to effectively inhibit DNA helicase activity, leading to bacterial genetic mutations or death.

#### 6.2.4. Rifamycin Class of Antibiotics [137]

The representative drugs of rifamycin class antibiotics are mainly fampicin and riforazide. Despite its efficacy against *C. trachomatis* and *C. pneumoniae* infection, rifampicin is not the first choice for treating Chlamydia infections. Riforazide exhibits potent anti-*C. trachomatis* activity and possesses remarkable cell penetration ability. Its mechanism of action involves binding to the β-subunit of Chlamydial DNA-dependent RNA polymerase, thereby inhibiting its activity and impeding the initial formation of the Chlamydia RNA chain, ultimately resulting in bactericidal effects. Importantly, these drugs exert minimal impact on human RNA polymerase, rendering them less harmful to humans.

#### 6.2.5. Aminoglycoside Antibiotic

The primary antimicrobial mechanism of aminoglycoside antibiotics against Chlamydia is to impede protein synthesis, rendering it a fast-acting bactericidal drug. In addition, aminoglycoside antibiotics can also enhance the impact of Chlamydia membrane proteins by inhibiting their synthesis, thereby affecting their barrier function and facilitating drug penetration into the cell. However, due to its limited ability to penetrate mammalian cells, its minimum inhibitory concentration (MIC) for anti-Chlamydia activity is considerably high at approximately 1 μg/mL [131].

#### 6.2.6. β-Lactam Antibiotics [138]

The MIC values of β-lactam antibiotics against Chlamydia are remarkably high, reaching 128 μg/mL and even exceeding 1024 μg/mL [131]. Nevertheless, these drugs exhibit reduced toxicity and are frequently used to treat Chlamydia infection in pregnant women. Amoxicillin is currently the primary choice for treating urogenital tract *C. trachomatis* infection in pregnant women. The anti-Chlamydia activity of these drugs relies on the presence of penicillin-binding proteins (PBPs) in bacteria, which play a crucial role in peptidoglycan metabolism through transpeptidase, carboxypeptidase, and endopeptidase activities. Three types of Chlamydia PBPs can be targeted by penicillin G. PBP1 exhibits transpeptidase activity while PBP3 displays carboxypeptidase activity. Carbapenems can bind to both PBP1 and PBP3, whereas cephalosporins and ampicillin can bind to PBP1 and PBP2. Mecillinam specifically targets PBP2. β-lactone ring antibiotics bind to PBPs, disrupting RBs division and resulting in persistent Chlamydia infection.

Despite Chlamydia remaining susceptible to all anti-chlamydial antibiotics and decades of treatment without confirmed in vitro resistance to first-line tetracyclines and macrolides in *C. trachomatis*, the bacterium continues to be widespread and notorious for its ability to cause long-lasting, persistent infections. One possible reason is that Chlamydia enters a resilient persistent state (RB state) in response to stress, characterized by non-cultivability under unfavorable conditions and a rapid return to normal replication once the stressor has been removed. Additionally, Chlamydia has evolved unique mechanisms to resist antimicrobial effects triggered by the host immune response, allowing it to cause long-lasting, persistent infections despite the wide availability of effective drugs [120].

### 6.3. Vaccine

The incomplete reliance on antibody response for host resistance to Chlamydia, combined with the ability of the obligate intracellular bacteria to employ multiple mechanisms in disrupting the immune response, has impeded the development of an effective vaccine for Chlamydia elimination [139,140]. The design of a vaccine for preventing chlamydial infection and disease involves identifying novel immunogen candidates, discussing effective delivery platforms and immunization routes, and choosing adjuvants to induce appropriate protective immunity at the targeted mucosal surfaces while minimizing severe inflammatory disease sequelae [141,142].

Chlamydia vaccine research has focused primarily on two vaccine antigen candidates, MOMP and CPAF [141]. Experimental studies with single-antigen vaccines have demonstrated that nMOMP induces the most robust protection against shedding and infertility. A promising alternative is a synthetic MOMP vaccine incorporating specific epitopes from various serovars [143]. Another approach is to develop a multivalent vaccine combining rMOMP with other well-conserved recombinant antigens, such as CPAF and/or Pmps [144]. To effectively prevent chlamydial infections, vaccines must elicit robust mucosal immune responses, which can be best achieved through sublingual, intranasal, and oral routes [141]. Additionally, effective and safe adjuvants, routes, and delivery systems are essential for inducing a robust immune response in the development of a *C. trachomatis* vaccine [143]. The research findings demonstrate that the utilization of TLR-derived adjuvants to target DCs with Chlamydia antigens effectively induces and amplifies the immune response of IFNγ-producing CD4^+^ T and CD8^+^ T cells, thereby exerting efficient control and prevention against *C. trachomatis* infection. This approach holds great promise for future development of Chlamydia vaccines [145].

## 7. Conclusions

In summary, as an obligate intracellular bacterium, Chlamydia’s unique biphasic reproductive cycle and reliance on host nutrients facilitate its infection and dissemination while presenting challenges for complete eradication. Although host cells employ various immune mechanisms to defend themselves, Chlamydia has developed multiple strategies to counteract these defenses and exacerbate disease progression by inducing host cell death. This makes it difficult to completely cure Chlamydia infection, and existing treatments mainly rely on antibiotics to alleviate symptoms, without an effective vaccine available. Therefore, conducting comprehensive research on the biology of Chlamydia and its interactions with its hosts is essential for developing more effective treatment and prevention strategies.

## Figures and Tables

**Figure 1 microorganisms-12-01302-f001:**
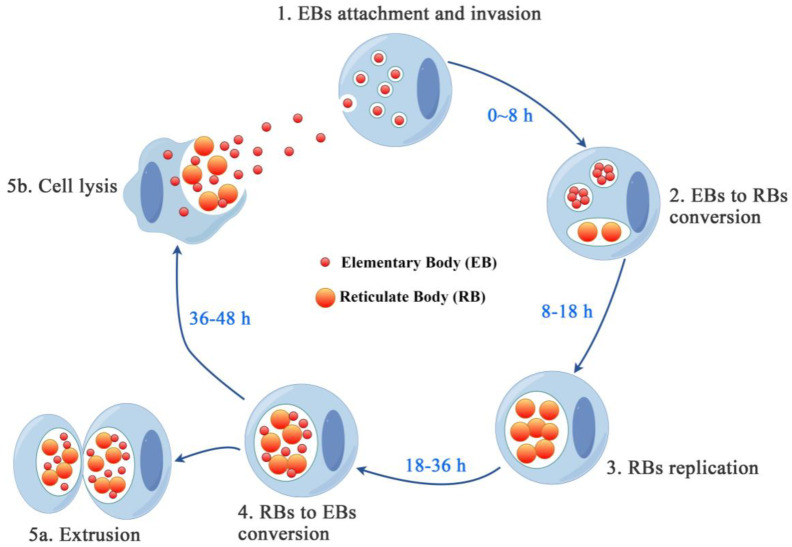
Chlamydia developmental cycle. The cycle begins with the attachment of infectious elementary bodies (EBs) to host cells, followed by their invasion into the cells to form vacuoles. Within 8–18 h, EBs differentiate into non-infectious reticulate bodies (RBs) and utilize host cell nutrients for replication. Within 18–36 h, RBs re-differentiate into EBs. Within 36–48 h, EBs are released either through host cell lysis or extrusion, infecting adjacent cells and initiating a new round of development. This figure was created using Figdraw.

**Figure 2 microorganisms-12-01302-f002:**
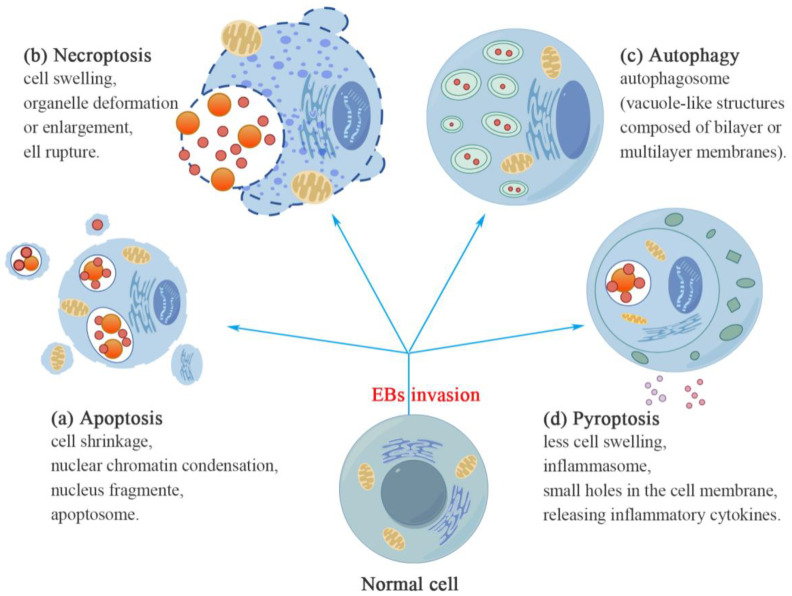
Host cell fates. Chlamydia-infected cells can undergo various cellular states, including programmed cell death (apoptosis), regulated necrosis (necroptosis), self-degradation process (autophagy), and inflammatory cell death (pyroptosis). (**a**) Apoptosis is characterized by reductions in cell size, condensation of nuclear chromatin, fragmentation of the nucleus, and presence of apoptosomes on the cell membrane surface. (**b**) Necroptosis is characterized by increased membrane permeability, which leads to cell swelling and organelle deformation or enlargement. The early nuclear morphology remains unchanged, followed by eventual cell rupture. (**c**) Autophagy is characterized by the presence of autophagosomes, which are vacuole-like structures composed of bilayer or multilayer membranes. (**d**) Pyroptosis is characterized by less cellular swelling, while Chlamydia infection induces the formation of an inflammasome within host cells, subsequently leading to the creation of small holes in the host cell membrane and releasing inflammatory cytokines. This figure was created using Figdraw.

## Data Availability

This review article examines the recent advancements in both domestic and international research. To obtain further information regarding this article, please get in touch with the author.

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
