# Peer review of "Insights into Chlamydia Development and Host Cells Response"

_microorganisms, 2024, doi:10.3390/microorganisms12071302_

Round 1
Reviewer 1 Report
Comments and Suggestions for Authors
This manuscript reviews the biphasic cycle development of Chlamydia and host cell responses during infection, as well as the acquisition of nutrients and intracellular metabolism of Chlamydia.
This manuscript is a good review that can be considered for publication. However, it requires significant corrections. One of them is that the authors consider two Chlamydia genres. It is regarded as one genre with 16 species and two candidates. Based on earlier reports from the literature. The Chlamydiaceae family currently consists of the single genus Chlamydia with 15 characterized species: C. abortus, C. avium, C. buteonis, C. caviae, C. felis, C. gallinacea, C. ibidis, C. muridarum, C. pecorum, C. pneumoniae, C. poikilothermis, C. psittaci, C. serpentis, C. suis, and C. trachomatis. In recent years, new candidate species were proposed based on molecular detection and identification methods, i.e., Candidatus C. corallus and C. sanzinia (Laroucau K, Ortega N, Vorimore F, Aaziz R, Mitura A, Szymanska-Czerwinska M, Cicerol M, Salinas J, Sachse K, Caro MR. Detection of a novel Chlamydia species in captive spur-thighed tortoises (Testudo graeca in southeastern Spain and proposal of Candidatus Chlamydia testudinis. Syst Appl Microbiol. 2020;43(2):126071. doi: 10.1016/j.syapm.2020.12607; Li Z, Liu P, Hou J, Xu G, Zhang J, Lei Y, Lou Z, Liang L, Wen Y, Zhou J. Detection of Chlamydia psittaci and Chlamydia ibidis in the Endangered Crested Ibis (Nipponia nippon).
Epidemiol Infect. 2020;148:e1. doi: 10.1017/S0950268819002231; ).
There are also many grammatical mistakes, and a redaction review is required. It is necessary for an English native-speaker reviewer. Some comments were added to the manuscript. Furthermore, it is essential to modify the conclusion because it does not have any relevant information about the review. It is necessary to exhaustively review the bibliography because some references, such as reference 3, do not correspond to the comments on the text. Furthermore, some references have only some authors and incomplete titles. It will help if you follow the author's instructions for citation references.

There are also many grammatical mistakes, and a redaction review is required. It is necessary for an English native-speaker reviewer.
Reviewer 2 Report
Comments and Suggestions for Authors
This review article summarizes the pathological mechanisms of chlamydia. The text is relatively easy to read and understand, but could benefit of more figures about the extracellular and intracellular molecules involved in the mechanisms.
Major comments:
(1) The authors could expand the description of the different types of disease associated with chlamydia infection, the pathogenesis, symptoms, and treatment.
a) Trachomatis
b) Psitacci
c) Pneumoniae
(2) The authors could describe the different types of drugs that are available in the market and explain what pathway the drugs are targeting.
(3) It may be beneficial if the genome characteristics of chlamydia are summarized.
(4) The article could include a description of the immunopathogenesis associated with the infection with chlamydia. How the host immune response attack chlamydia and how chlamydia modulates the immune response to evade it.
a) IFN gamma
b) cell-autonomous immunity
c) innate immune response
d) adaptive immune response
Please refer, for example:
Helble JD, Starnbach MN. T cell responses to Chlamydia. Pathog Dis. 2021;79(4):ftab014. doi:10.1093/femspd/ftab014 Dockterman J, Coers J. Immunopathogenesis of genital Chlamydia infection: insights from mouse models. Pathog Dis. 2021;79(4):ftab012. doi:10.1093/femspd/ftab012 Dockterman J, Reitano JR, Everitt JI, et al. Irgm proteins attenuate inflammatory disease in mouse models of genital Chlamydia infection. mBio. 2024;15(4):e0030324. doi:10.1128/mbio.00303-24 Zhong G. Chlamydia Spreading from the Genital Tract to the Gastrointestinal Tract - A Two-Hit Hypothesis. Trends Microbiol. 2018;26(7):611-623. doi:10.1016/j.tim.2017.12.002Ma Y, Sun J, Che G, Cheng H. Systematic Infection of Chlamydia Pneumoniae. Clin Lab. 2022;68(8):10.7754/Clin.Lab.2021.210908. doi:10.7754/Clin.Lab.2021.210908 Mishori R, McClaskey EL, WinklerPrins VJ. Chlamydia trachomatis infections: screening, diagnosis, and management. Am Fam Physician. 2012;86(12):1127-1132. Murray SM, McKay PF. Chlamydia trachomatis: Cell biology, immunology and vaccination. Vaccine. 2021;39(22):2965-2975. doi:10.1016/j.vaccine.2021.03.043 (5) You may comment about vaccination. Rey-Ladino J, Ross AG, Cripps AW. Immunity, immunopathology, and human vaccine development against sexually transmitted Chlamydia trachomatis. Hum Vaccin Immunother. 2014;10(9):2664-2673. doi:10.4161/hv.29683 Liang S, Bulir D, Kaushic C, Mahony J. Considerations for the rational design of a Chlamydia vaccine. Hum Vaccin Immunother. 2017;13(4):831-835. doi:10.1080/21645515.2016.1252886 de la Maza LM, Zhong G, Brunham RC. Update on Chlamydia trachomatis Vaccinology. Clin Vaccine Immunol. 2017;24(4):e00543-16. Published 2017 Apr 5. doi:10.1128/CVI.00543-16 Cochrane M, Armitage CW, O'Meara CP, Beagley KW. Towards a Chlamydia trachomatis vaccine: how close are we?. Future Microbiol. 2010;5(12):1833-1856. doi:10.2217/fmb.10.148
(6) This is a very interesting review.
Elwell C, Mirrashidi K, Engel J. Chlamydia cell biology and pathogenesis. Nat Rev Microbiol. 2016;14(6):385-400. doi:10.1038/nrmicro.2016.30 Andersen SE, Bulman LM, Steiert B, Faris R, Weber MM. Got mutants? How advances in chlamydial genetics have furthered the study of effector proteins. Pathog Dis. 2021;79(2):ftaa078. doi:10.1093/femspd/ftaa078
Minor comments:
(4) Line 72, please correct "dEBsated".
(5) Line 77, please correct "Sturd1".
(6) Line 85, please correct "therEBsy"
(7) Line 93, please correct "therEBsy"
(8) Line 150, please correct "Ttranslocated"
(9) Section 5 explains that host cells defend against chlamydia by the apoptosis pathways (and different variants). However, in 5.2 section, the pro-survival effects on host cells by chlamydia are described.
"5.1. Cell fates" vs. "5.2. Anti-death strategies of host cells". Does the host cells induce apoptosis to defend against infection, and chlamydia prevent apoptosis/cell death to multiply inside the host?
English is fine. I think the text could be polished using English editing software.
Round 2
Reviewer 2 Report
Comments and Suggestions for Authors
thank you for the revision
Comments on the Quality of English Languagethere were mistakes but with editing should be fine